# Assessment of the Impact of Pentafecta Parameters Affecting the Quality of Life of Patients Undergoing Laparoscopic Radical Prostatectomy

**DOI:** 10.3390/ijerph20020944

**Published:** 2023-01-04

**Authors:** Mateusz Wojtarowicz, Adam Przepiera, Artur Lemiński, Adam Gołąb, Marcin Słojewski

**Affiliations:** Department of Urology and Urological Oncology, Pomeranian Medical University, Powstańców Wielkopolskich 72, 70-111 Szczecin, Poland

**Keywords:** prostate cancer, laparoscopy, pentafecta, quality of life

## Abstract

Prostate cancer is being detected in increasingly younger men. These patients expect to preserve their current quality of life and quickly recover after treatment. Medical technology and surgical techniques are advancing along with the growing expectations of patients. In addition, the universal method of assessing the quality of outcomes after operations is constantly being researched. As of today, biochemical remission alone, after radical prostatectomy, is insufficient for the patient. Therefore, multi-parametric evaluation methods are being developed, such as trifecta, which assesses biochemical remission, continence, and erectile function. The improvement over the trifecta is the pentafecta, which additionally evaluates postoperative complications and infiltration of surgical margins. Our study was conducted within a group of patients who were surgically treated for prostate cancer in 2017 at the Clinic of Urology and Urological Oncology of the Pomeranian Medical University. We recruited 237 men for the study. From that group, 131 men met the criteria to be included in the analysis. Maintaining continence (87.78%) is the easiest pentafecta parameter to obtain and will have the greatest impact on quality of life in the future. Maintaining biochemical remission (82.44%) is the second most important aspect for the patient. Retaining erectile function is the most difficult pentafecta parameter to obtain (29.01%) while having little impact on the quality of life. Negative surgical margins (66.41%) showed a negligible impact on the quality of life. The occurrence of complications (32.07%) has a negative impact on the quality of life of patients, but only during the treatment of complications.

## 1. Introduction

Thanks to numerous social campaigns and the efforts of the urological community, men’s awareness of prostate cancer and the need for early diagnosis is increasing. More and more men are presenting for prophylactic urological examinations, and since the association of PSA levels with the occurrence of prostate cancer, the diagnosis of this disease has increased significantly. Common access to PSA testing means that prostate cancer is diagnosed in younger men. On the other hand, the diagnosis of clinically insignificant diseases may lead to overtreatment. However, long-term observations confirm that radical treatment is preferable to careful observation because it increases overall survival, especially in younger patients from the intermediate- and high-risk groups [1,2]. That is why it is so important to diagnose the disease early, to properly assess the risk associated with it, and to adjust the optimal therapeutic treatment. In some cases of low-risk neoplasms, active surveillance can be offered to the patient, although the criteria for its inclusion have not yet been clearly established. It is estimated that 2–45% of patients who can be qualified for active surveillance will require radical treatment in the future. Currently, radical prostatectomy or radiotherapy is used as the basis for the treatment of prostate cancer, taking into account various types of these techniques. There are no randomized multicentre studies comparing these two forms of treatment; however, on the basis of the single-center and multicentre studies presented in the latest publications, assessing the survival of each method, it can be concluded that their effectiveness is comparable.

The progress of medical technology affects the expectations of patients. They want the treatment not only to be effective but also not to affect their current quality of life. Therefore, a simple system for evaluating the effects of surgical treatment of prostate cancer, described as Trifecta, was introduced. The Trifecta system adopted so far (oncological remission, urinary continence, and erectile function) lacked parameters directly related to the perioperative period. Therefore, Patel proposed adding negative surgical margins and the assessment of perioperative complications to the above.

Positive surgical margins as a result of histopathological examination cause anxiety in the patient. They are afraid of non-radical treatments. This is due to the fact that the decision on further adjuvant treatment should be made on the basis of the PSA assessment after surgery. Due to the 2-week half-life of PSA, we usually evaluate the first PSA test after 6 weeks. During this time, patients who have positive margins experience uncertainty and associated stress. Therefore, it is a parameter that affects the quality of life.

## 2. Materials and Methods

The study was designed as a prospective, non-randomized, single-center, open-label, non-controlled study. It was decided to compare the obtained results with the data published in research papers on the results of surgical treatment of prostate cancer, with particular emphasis on publications on laparoscopic radical prostatectomy. Information was obtained from patients using the validated IIEF-5 questionnaires (International Index of Erectile Function), IPSS (International Prostate Symptom Score), the Clavien–Dindo scale, EPIC—urinary tract (Expanded Prostate Cancer Index Composite), and the in-house questionnaires prepared for the study: assessment of operating conditions by the operator (Appendix A), assessment of the quality of vesicourethral anastomosis according to the internal classification system by prof. dr. hab. n. med. Marcin Słojewski (Appendix B), and assessment of the quality of life in the subjective perception of the patient on a scale of 1–6 (Appendix C). The burden of comorbidities was assessed according to the Charlson scale. Observations were first made on enrollment day 0, then on day 2, discharge day, week 6, and months 3, 6, and 12. The preferred form of contact was meeting the patient during the follow-up visits. If it was impossible to contact them in person, the patients were asked to send questionnaires by e-mail, or telephone conversations were held with them. In addition, 237 patients of the Clinic of Urology and Urological Oncology with histopathologically confirmed prostate cancer who underwent laparoscopic radical prostatectomy between 1 January 2017 and 31 December 2017 were invited to participate in the study.

All invited patients agreed to participate in the study. They signed the informed consent form after reading the information sheet and talking to the doctor. Twenty people withdrew from participating in the study during its duration. A Complete follow-up was obtained for 217 patients.

The following inclusion criteria for the study were adopted:−Sexually active men, who scored at least 12 points in the IIEF-5 male questionnaire;−Qualifying for the procedure with the sparing of nerve-nerve bundles by the urologist performing the surgery. Qualification was performed in accordance with the guidelines of the European Society of Urology and was undertaken on the basis of clinical data from a physical examination and radiological examinations.

The criteria for excluding the patient from the study were confirmed urinary incontinence before surgery and loss of contact with the patient, which made it impossible to obtain a complete follow-up. Each patient had the right to withdraw from the study at any time. There were 131 men who met the above inclusion criteria for the study. Earlier urological procedures on the lower urinary tract did not disqualify patients from the study.

The demographic, oncological, and functional characteristics are presented below (Table 1 and Table 2).

Oncological evaluation after laparoscopic radical prostatectomy was performed in accordance with the recommendations of the European Society of Urology. The PSA concentration value of 0.2 ng/mL was used as the biochemical recurrence threshold [3]. The tests were performed using the ultrasensitive serum PSA level assessment technology. The accuracy threshold of PSA concentration measurement for this method is up to 0.008 ng/mL [4]. The first postoperative measurement was performed 6 weeks after the surgery. According to Stamey and McNeal, after such a period, the serum PSA concentration should be undetectable [5]. Subsequent measurements were carried out in the 3rd, 6th, and 12th months after the procedure, according to the generally accepted scheme [6]. The criterion of a positive or negative surgical margin was assessed on the basis of the result of the histopathological examination, performed in all cases by an experienced histopathologist (Department of Pathomorphology SPSK2 PUM, head of the department: prof. Dr hab. N. Med. Jan Lubiński). The report from this study was described in accordance with the recommendations of the European Society of Urology and the International Association of Urological Pathologists (ISUP) below [7,8] (Figure 1).

Functional assessment was divided into the domains of sexual function and continence. The validated IIEF (International Index of Erectile Function) questionnaire, version 5, was used to assess sexual function and compare it to the baseline state before radical prostatectomy [9,10]. The possibility of having sexual intercourse with the use of phosphodiesterase-5 inhibitors or without the need to use phosphodiesterase-5 inhibitors was assumed to meet the criterion of the return of sexual function and was assessed by the patient as satisfactory (in the IIEF-5 questionnaire, it is defined as “Moderate—3 points”). The evaluation of urinary continence was performed using the validated EPIC questionnaire (Expanded Prostate Cancer Index Composite—urinary tract) [8] and the number of hygienic pads used by the patient. Quantity evaluation, or summing up the daily weight of the used hygienic liners, is a method commonly adopted by researchers, and its values strongly correlate with the results of the video urodynamic test and questionnaires [1,2]. The criterion of full continuity was met when there was no need for the patient to use hygienic pads or use one pad per day. In this case, the weight of urine lost within 24 h could not exceed 4 g. Postoperative complications were assessed twice. The first examination, carried out on the day of discharge from the hospital, assessed early complications. The second examination, carried out six weeks after the operation, assessed late complications. In both cases, the validated Clavien–Dindo scale was used [11].

Statistical analysis was carried out using the R version 3.6.0 statistical software from “The R Foundation for Statistical Computing”. The following measures of location and dispersion were used to assess demographic, oncological, and clinical parameters: mean and standard deviation (SD) for small numbers (less than 100 per group) and for all numbers with non-normal or atypical distribution parameters. The median, median absolute deviation (MAD), and interquartile range were used for numbers with a normal distribution parameter. Multiparameter analysis was performed using Fisher’s exact test. An Anova Repeated Measures test was performed, and for non-parametric distribution, an Aligned Rank Transform was used. Benjamini–Yekutieli (False Discovery Rate) corrections were made everywhere. Then, logistic and linear regression were performed using the Bonferroni correction, and OR was taken as the measure of the effect size. For repeatable categorical measurements, a marginal homogeneity test was performed. These tests check whether there is a difference in groups over time—for categorical variables. The level of statistical significance in single-parameter and multi-parameter Fisher’s tests was set at *p* ≤ 0.05. The level of statistical significance for the Bonferroni correction was set at *p* < 0.001.

The one-parameter analysis showed that there is a statistically significant (*p* ≤ 0.05) relationship between the occurrence of pentafecta and the result of histopathological examination—histopathological stage expressed in the TNM scale (*p* = 0.048) and the degree of histopathological architecture disorder Gleason/ISUP (*p* = 0.004). Parameters that came close to statistical significance were the presence of postoperative leukocytosis (*p* = 0.088), patient’s age (*p* = 0.01), preoperative Gleason/ISUP sum (*p* = 0.110), and operator experience at the time of surgery (*p*= 0.118). The demographic factors that were included in the analysis and with which the pentafecta score did not correlate were: BMI, waiting time for surgery from the time of the biopsy, prostate volume, IIEF-5 score, and presence of comorbidities. There was also no correlation between pentafecta and the following oncological parameters included in the analysis: percentage of occupied biopsy specimens, PSA concentration at the time of diagnosis, NMR examination of the prostatic capsule, and preoperative d’Amico risk group. Among the intraoperative parameters taken into account, with which the pentafecta result showed no correlation, there were: the quality of the anastomosis, intraoperative blood loss, duration of the procedure, intraoperative amount of adipose tissue, anatomical conditions within the pelvis, difficulty in tissue dissection, local difficulties in removing the prostate, difficulties in reconstructing the urinary tract, and the number of removed lymph nodes. The postoperative parameters included in the analysis, which did not affect the pentafecta value, included an increase in creatinine, a decrease in hemoglobin, a leak from the drain, erection rehabilitation, and antibiotic prophylaxis. 

In the multiparametric analysis, the postoperative Gleason/ISUP grade (*p* = 0.005) and local difficulties in prostatectomy (*p* = 0.030) were indicated as independent factors predisposing to obtaining pentafecta. Operator experience (*p* = 0.060) and prostate volume (*p* = 0.081) were located in close proximity to statistical significance. After the Bonferroni correction, none of the parameters showed statistical significance (the *p* cut-off for the Bonferroni correction was 0.001).

## 3. Results

Before the operation, patients filled out a questionnaire assessing their quality of life. After the prostatectomy, the assessment of the quality of life was performed again at each of the scheduled control visits. Then, the individual components of the pentafecta and their absence were compared with the patient’s self-reported subjective assessment of the quality of life after surgery. Not all patients who obtained pentafecta assessed their quality of life after the surgery as having the maximum number of points. There were 14/19 (73.68%) patients were fully satisfied (they scored 6/6 points) and reported no complaints within twelve months after the surgery. Additionally, 5/19 patients (26.32%) rated their quality of life at 5/6 points and reported minor reservations: two used one sleeper a day—“just to be sure”, two used PDE-5 inhibitors to obtain an erection, and one of the patients was concerned about the necessity of periodic PSA examinations. The average quality of life value was pairwise stratified with the pentafecta component. In this way, a picture of the influence of the ailments on the quality of life was obtained. Based on the information obtained, a hierarchy of the weights of individual components was prepared. 

Complications require special interpretation. Most of the complications that occurred were minor—Grades 1 and 2 on the Clavien–Dindo scale. Serious complications—Grade 3a in the Clavien–Dindo scale and above—were uncommon. No patient died or required treatment in the intensive care unit. At the time of the complication and in the weeks immediately following, they were the most important for the patient. With recovery, however, the complications lost their impact on quality of life. The values are presented in the tables below (Table 3 and Table 4). The hierarchies of the influence of pentafecta parameters on the quality of life are presented in the form of figures (Figure 2). 

## 4. Discussion

The idea behind introducing pentafecta was to meet the increasing expectations of patients. Patel, proposing the pentafecta criteria, pointed out that 13.2% of patients who obtained trifecta would not obtain pentafecta due to positive surgical margins or the occurrence of complications [9]. In this study, the difference between trifecta and pentafecta was 11.45%. According to Patel’s hypothesis, it determines the percentage of patients who may be dissatisfied with the operation despite meeting the trifecta.

Analyzing the results of our study, we noticed that the occurrence of complications significantly reduces the quality of life compared to the classic aspects of pentafecta. 

The presence of positive surgical margins did not show a significant effect in this respect. Bourke et al. noted that the deterioration of quality of life accompanies all methods of prostate cancer treatment, not only surgery but also radiotherapy and hormone therapy [12]. The authors cite data showing that the deterioration of physical and mental well-being occurs as a result of the treatment of all neoplastic diseases, not only prostate cancer. According to the authors, the support of the patient during and after the therapy is very important in minimizing the side effects of treatment, and the best results were achieved by those patients who worked according to individually tailored recommendations. This conclusion is related to the previously cited work by Saloni, in which the author draws attention to the diversity of patients’ expectations [13]. Schroeck, in his study of the quality of life of patients after prostatectomy, which concerned the preoperative expectations of patients, noted that the expectations of those who were treated with robot-assisted surgery were higher [14]. Patients completed the questionnaires during qualification for treatment. Despite the lack of differences in age, workload, and PSA level, the group of patients who qualified for robot-assisted prostatectomy expected a shorter hospital stay and a faster return to physical activity. Summarizing both studies, one on preoperative expectations and the other on the quality of life after surgery, Schroeck stated that with the development of medical technologies, expectations increased and the tolerance to complications decreased among patients. 

The modern patient wants to actively participate in the choice of treatment method. Obtaining a good oncological outcome is most important but achieving it at the price of a significant deterioration of the quality of life is often unacceptable for patients. To meet their expectations, we must constantly improve the technique of surgery and precisely qualify patients for treatment. Pentafecta is a very accurate method of assessing the quality of a radical prostatectomy. Its detailed analysis allows for the identification of components that require improvement. Preservation of sexual function (29.01% of our material) is the most difficult element of pentafecta to obtain. Leaving undamaged neurovascular bundles has the greatest influence on the preservation of erection. Operative difficulties due to oncological advancement are the main factor responsible for postoperative erectile dysfunction. Preservation of continence (87.78%) is the easiest pentafecta parameter to obtain; it also has the greatest impact on the quality of life in the long term. It was noticed that the intraoperative difficulties in tissue preparation and restoration of the continuity of the urinary tract were the most determinant of continence maintenance. Negative surgical margins (66.41%) and biochemical remission (82.44%) confirmed their strong correlation. Both postoperative oncological parameters are determined to the greatest extent by the result of the histopathological examination. The degree of obesity and intraoperative difficulties with the reconstruction of the urinary tract have a significant influence on the occurrence of complications (32.07%).

## 5. Conclusions

Preservation of continence (87.78%) is the easiest to obtain parameter of pentafecta and, at the same time, has the greatest impact on the quality of life in the long term. Maintaining biochemical remission (82.44%) is the second-most important parameter. Maintaining sexual performance is the most difficult pentafecta parameter to obtain (29.01%), at the same time having little impact on the quality of life. Negative surgical margins (66.41%) show little effect on the quality of life. The occurrence of complications (32.07%) has a negative impact on the patient’s quality of life.

The impact of treatment on quality of life is becoming an increasingly important factor when choosing a method. The dynamic development of minimally invasive techniques as well as non-surgical treatment techniques such as radiotherapy is aimed at a quick recovery for patients. The described study showed that not all postoperative parameters have an equal impact on the patient’s quality of life. The first results of observations suggest that the effects of everyday functioning may be slightly more important than the final oncological effect. This is very important information for urologists. This issue requires further investigation. Long-term observation is necessary with the assessment of patients’ changing expectations with their age. 

## Figures and Tables

**Figure 1 ijerph-20-00944-f001:**
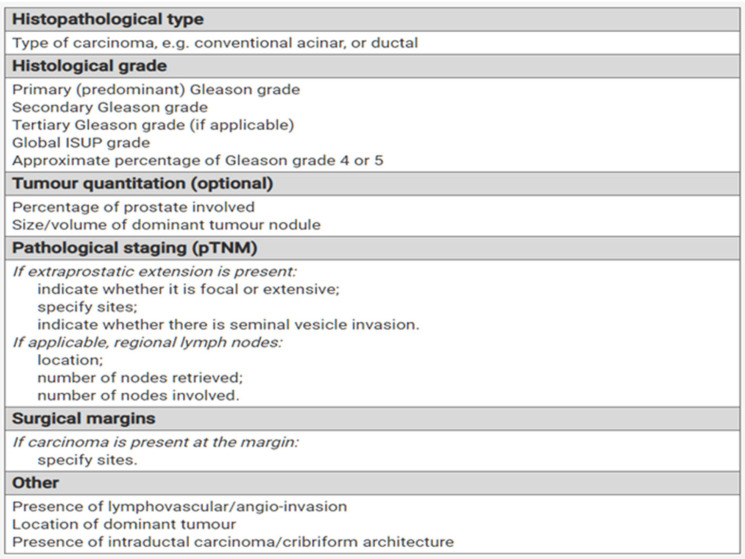
International Society of Urological Pathology (ISUP) 2014 grade (group) system.

**Figure 2 ijerph-20-00944-f002:**
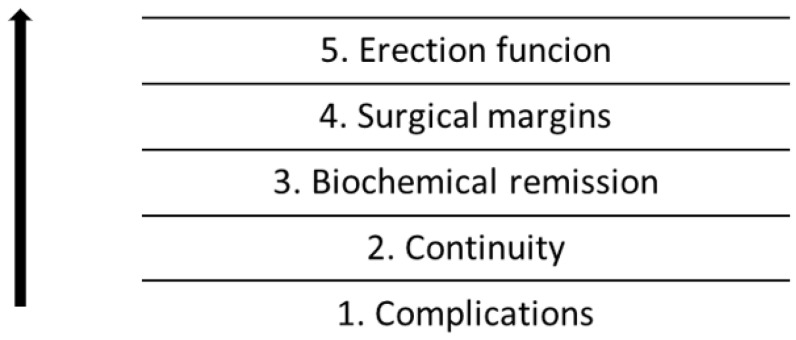
Hierarchy of parameters in relation to the quality of life, from the most important (1) to the least important (5).

**Table 1 ijerph-20-00944-t001:** Demographic characteristics of the study group.

Characteristic	Results (*n* = 131)
Number of patients recruited	237
Number of patients followed	217
Number of patients meeting the inclusion criteria for the study	131
Age, mean (SD)	62.83 (±5.43)
BMI, mean (SD)	27.80 (±3.44)
Charlson scale, median	2 (0–3)
Positive family history (%)	10 (7.63%)
Education:	
Basic, vocational (%)	30 (22.90%)
Technical secondary (%)	59 (45.03%)
Higher, Bachelor (%)	42 (32.06%)

**Table 2 ijerph-20-00944-t002:** Oncological and functional characteristics of the study group.

Characteristic	Results (*n* = 131)	Characteristic	Results (*n* = 131)
PSA, mean (SD)	9.43 (±5.96)	Gleason/ISUP:	
PSA, median (min-maks)	7.6 (3.2–43.6)	1 (%)	90 (68.70%)
Clinical advancement:		2 (%)	27 (20.61%)
cT1c (%)	48 (36.64%)	3 (%)	8 (6.10%)
cT2a (%)	27 (20.61%)	4 (%)	6 (4.58%)
cT2b (%)	38 (29.00%)	Risk group d’Amico:	
cT2c (%)	15 (11.45%)	Low	38 (29.00%)
cT3a (%)	1 (0.76%)	Intermediate	65 (49.61%)
cT3b (%)	2 (1.52)	High	28 (21.37%)
Amount of material seized, median (Extremes)	42% (8–75%)	Waiting time from biopsy to prostatectomy in days, mean (SD)	138.79 (±190.61)
IIEF-5, mean (SD)	20.57 (±3.99)	IIEF-5, median (Ekstrema)	21 (18–25)

**Table 3 ijerph-20-00944-t003:** List of unfulfilled pentafect components mentioned by the patient as the most inconvenient (cases *n* = 131).

Component of Pentafecta		6 Weeks	3 Months	6 Months	12 Months
Erectile function		51	65	63	57
Urinary continence		50	37	26	26
Surgical margins		3	2	1	1
Biochemical remission		14	18	21	25
Complications		8	6	4	3
Other		3	1	2	2
No complaints		2	2	14	17

**Table 4 ijerph-20-00944-t004:** Summary of the average declared quality of life in patients reporting the indicated ailment as dominant (scale of 1–6, Appendix A).

Component of Pentafecta		6 Weeks	3 Months	6 Months	12 Months
Erectile function		4.52	4.78	4.95	4.96
Urinary continence		3.82	4.27	4.19	4.38
Surgical margins		4.33	4	5	5
Biochemical remission		4.14	4.38	4.38	4.48
Complications		3.5	4	4.5	5.33
Other		4.33	4	3.5	4
No complaints		5.5	6	5.5	5.74

## Data Availability

https://nauka-polska.pl/#/profile/research?id=351945&_k=4q017n accessed on 30 September 2022.

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
