# Peer review of "Assessment of the Impact of Pentafecta Parameters Affecting the Quality of Life of Patients Undergoing Laparoscopic Radical Prostatectomy"

_ijerph, 2023, doi:10.3390/ijerph20020944_

Round 1
Reviewer 1 Report
The authors assessed the impact of pentafecta parameters affecting the Quality of Life of patients undergoing Laparoscopic Radical Prostatectomy. Interesting topic but poorly presented. Please perform some statistical calculations to show the significance of your results. Please include figure legends.
Author Response
Thank you very much for your evaluation. I am glad that you pointed out the factors that I omitted. I have included the answers to your comments in the attachment

Reviewer 2 Report
The authors have done a good job in a prospective, non-randomised, single-center, open-label, non-controlled study. The manuscript is well written, and brings novelty to this field of urology. They present a good overview of quality of life in intubated patients.
However, the authors should carry out a grammatical revision to correct small errors throughout the manuscript.
Author Response
I am very grateful for the time I spent to make this work better. We reviewed the work once again in terms of grammar and vocabulary. We have made corrections to make the work more readable.
Reviewer 3 Report
How do these margins affect quality of life? I understand that there may be 2 issues at stake - the stress of "leaving" the cancer and the need for complementary radiation therapy, so again, stress and long treatment and complications of that treatment. Observation time short, so late complications of radiotherapy are not observed.
Please add if and how many patients received radiotherapy after surgery. Were there any differences in quality of life between men who were irradiated and those who were not.
Please describe in more detail what "complications" are hidden under this term.
It is worth mentioning that the follow-up time is short and, in fact, the article is about early quality of life after surgery.
Figure 2 not very pertinent, could be reworked into some more pictorial one.
In table 3 AND 4 necessary to add description - not sure if cases or %.
In the summary necessary some sentence for the future , because there only a summary of the results.
Author Response
I am very grateful for the time I spent to make this work better. Your comments were very helpful. I have included the answers in the attachment.
